# How the 5G Enabled the COVID-19 Pandemic Prevention and Control: Materiality, Affordance, and (De-)Spatialization

**DOI:** 10.3390/ijerph19158965

**Published:** 2022-07-23

**Authors:** Gaoyong Li, Xin Zhang, Ge Zhang

**Affiliations:** School of Management Science and Engineering, Shandong University of Finance and Economics, Jinan 250014, China; ligaoyong0706@163.com (G.L.); zhangxin@sdufe.edu.cn (X.Z.)

**Keywords:** 5G, COVID-19 pandemic, materiality, affordance, spatialization and de-spatialization

## Abstract

5G, the most disruptive innovation, had played a significant role in the COVID-19 pandemic prevention and control. However, as a novel technology and context, we have little knowledge about how 5G enabled pandemic prevention and control. This study collected 212 cases and conducted qualitative research to explore how the 5G worked in prevention and control. Based on the concepts of materiality and affordance, we grounded two affordances of spatialization and de-spatialization from the data. Spatialization provides non-contact ways to complete the tasks which are supposed to be completed in contact, and de-spatialization provides remote operations to complete the tasks which are supposed to be completed on-site. Spatialization and de-spatialization enabled the diagnosis and treatment of the infectors to relieve the unbalance of medical staff, cutting the infectious route to contain the viral spread, and logistic supply to support the prevention and control. Our study offers theoretical contributions to digital pandemic prevention and control, and the literature on 5G also offers practical implications.

## 1. Introduction

The COVID-19 pandemic broke out in Wuhan, China, in early 2020, and soon spread all over the world. China’s central and local governments acted with the strictest public health intervention such as implementing lockdown of the provinces, cities, and even communities to prevent and control the pandemic. It took approximately two and a half months that the routine life and external transportation in Wuhan, the city with the most severe epidemic, resumed to normal, which meant the large-scale spread of COVID-19 pandemic in China had been contained. In this process, digital technologies including big data analytics (BDA), artificial intelligence (AI), and 5th generation communication (5G) greatly improved the efficiency of pandemic prevention and control by enabling medical care about the infectors, tracing individual travel history, and community management. As a novel technology, 5G had a played critical role. For example, the outbreak of the epidemic caused the inadequacy of medical staff and resources in the severe areas, 5G-based telemedicine assisted to diagnose and treat the infectors; 5G-based non-contact body temperature measurements screened the persons with an abnormal temperature in fast-flowing crowd to exclude the potential infectors.

5G technology as a new generation of communication has the characteristics of enhanced mobile broadband (eMBB), ultra-reliable and low latency communication (URLLC), and massive machine-type communications (mMTC) [1]. However, its emergence was rapid, which caused its application scenarios to not be fully understood [2]. Therefore, extant research on 5G mainly focuses on its technological aspects, such as the deployment of the 5G network and there is little research on its social and business values [2]. We retrieved the Basket of Eight journals and other important journals in the IS field such as I & M, DSS, and so on, and found that there was little literature whose research target or question was 5G and its influence.

However, the significant role that 5G played in the COVID-19 pandemic prevention and control had attracted the attention of scholars [3]. They explored how 5G assisted the fight against COVID-19, for example, 5G-based ultrasound can assist to detect infected patients [4]. Bluetooth low energy that is connected directly with 5G network can facilitate contact tracing [5]. These studies can be classified into two streams: (1) the first stream was from scholars who majored in the medical field, so they viewed 5G as a tool which supported medical diagnosis and treatment (e.g., [6,7]), (2) the second stream presented the specific context or case in which 5G helped to combat COVID-19 (e.g., [5]). These two streams offered little theoretical insight on how 5G enabled pandemic prevention and control. The fact that 5G played important roles in the battle against COVID-19 offers us an opportunity to observe and theorize how 5G enabled pandemic prevention and control to attain insight into this brand new and influential digital innovation.

Hence, we adopted this context to empirically ground the theoretical understanding that attends to the materiality of 5G and how the materiality enables the pandemic control. Our study has three objectives: (1) to develop an empirical description of how 5G has been used to prevent and control the pandemic, (2) to identify the pertinent material features of 5G that facilitated the control of the pandemic, and (3) to integrate these findings into an empirically grounded theoretical model of 5G-enabled pandemic prevention and control. Employing affordance theory, we theorized the ways that 5G enabled pandemic prevention and control as spatialization and de-spatialization. The former means that 5G can help to complete tasks in non-contact way which had to be completed by close contact before; the latter means that 5G can help to eliminate the problems that are brought by the geographic space to accomplish the tasks that have to be accomplished on the spot.

## 2. Theoretical Background

### 2.1. Pandemic Control

Since the birth of human beings, infectious diseases have accompanied us [8]. Last century had witnessed large-scale outbreaks of infectious diseases very often, such as the 1918 Spanish influenza [9], Asian flu pandemic during 1957–1958 [10], Hong Kong flu pandemic during 1968–1969 [11], the SARS in 2003 [12], H1N1 Pandemic [13], and the MERS in 2011 [14], and the time elapse between two infectious diseases became shorter. In addition to pandemics, some diseases also spread in local areas, causing very serious consequences. For example, the outbreak of Ebola virus in the Congo caused a large number of deaths every time [15]. With the progress of science and technology, human beings have made great progress in the prevention and control of pandemics [16]. Vaccines are the most effective way to form group immunity, thus blocking the spread of the virus [17]. However, due to the lagging development of vaccines and the rapid variation of some viruses, it is hard to prevent and control epidemics by vaccines [18]. 

Therefore, public management such as detecting, quarantining, and tracking infectors have become the main means of prevention and control of pandemics, which can be illustrated by the lockdown of Wuhan [19]. How to control the pandemic is not only a medical problem and is more of an onerous challenge of public management [20]. From the angle of public management, the main tasks of infectious disease control include detection, prevention, response, recovery, and management of index cases [21]. WHO’s instructions on the prevention of COVID-19 included washing hands with soap and water, keeping social distance, and practicing respiratory hygiene [22]. Based on the extent of the spread of COVID-19, China CDC made appropriate policies and measures, such as locking down the city, conducting closed management of communities, and wearing masks. All these measures proved to be effective and efficient in COVID-19 control [23,24].

However, with the social system becoming increasingly complex, complex system theory shows that the more complex the system is, the lower stability it is. Therefore, strict public health measures are subject to restrictions. The emergence of digital technology has alleviated this limitation to a certain extent [25]. AI technology enabled the pandemic prevention and control [26] by predicting the spread of the disease [27] and assisting to screen out medicine which can cure the COVID-19 from existing drugs [28,29]. Telemedicine was used to cure the infectors while avoiding exposure and viral transmission [30]. All of them digitalized the prevention and control of the pandemic.

### 2.2. The Fifth-Generation Communication Technology (5G)

5G is a new generation communication technology, which is viewed as the most disruptive innovation [31]. The most salient features of 5G are enhanced mobile broadband (eMBB), which means a huge volume of data can be transmitted at very high speed; ultra-reliable and low latency communication (URLLC), which means transmission of small payloads with very high reliability from a limited set of terminals; and massive machine-type communications (mMTC), which means diverse and numerous devices can connect into the same network at the same time without mutual interference [1,32].

The unique traits of 5G make it be the enabler of digital technologies such as IoT, BDA, cloud, and SDN [33], and how 5G enables digital technologies is illustrated in Figure 1. 5G offers a network with a high bandwidth and ultra-low latency for IoT applications so that the data that are generated by IoT application can be delivered more efficiently and economically, and these data can become the resources of big data analytics. 5G radio access network can turn the cloud into cloud-based RAN, which enables the data centers to balance the load and resource allocation to achieve smart routing, better traffic management, and to improve network resource utilization.

Based on the integration of 5G and other digital technologies, lots of applications have been visioned and piloted and some of them have been applied on a small-scale. Taking telesurgery as an example, the remote operation site where an operator controls the slave is connected to the on-site surgery site where a slave manipulator performs on the patient by the network site. In this process, ultra-reliability and ultra-low latency of the network are critical, however, the characteristics of the extant networks hardly meet these requirements. 5G’s features of eMBB, URLLC, and mMTC solve these problems and make telesurgery feasible. Table 1 offers more application scenarios of 5G.

However, 5G technology is still in its fledgling stage [38] and hasn’t been commercialized universally [33]. Most of the application scenarios that are presented in Table 1 are prognostic and in the experimental phase [48]. While these studies mainly focus on the engineering and technical requirements of the application scenarios, little research focuses on the business and social value of 5G [48,49]. However, the research on the business and social value of 5G has found that the impact of mobile broadband services that are provided by the 5G network on the business value is positive [49]. Especially, 5G can play an important role in crisis management, for example, 5G is more helpful for real-time dissemination of emergency warning messages than conventional short-range communication [50].

The outbreak of COVID-19 offers us an opportunity to observe what influence 5G exerts in this crisis. The related research can be categorized into how 5G facilitated pandemic control and how to eliminate the rumors of 5G and the panic that was entailed by the rumors. The former explores how 5G enabled the fight against COVID-19, such as robot-assisted remote ultrasound system [51], e-health [52], telemedicine [53], and so on. The latter is about the negative effect of 5G. During the COVID-19 pandemic, a conspiracy theory about 5G had prevailed around some countries, and scholars used data from social media to study why the conspiracy theory happened and spread, and how to annihilate the conspiracy theory (e.g., [54]).

Through the analysis of the literature on 5G and its impact on pandemic prevention and control, it can be found that research on 5G mainly focuses on the possible application scenarios and their engineering requirements, neglecting its business and social value. Studies on how 5G assisted in combating the COVID-19 pandemic mainly centered on the technical facet of telemedicine or presented cases which could not offer theoretical insight. Therefore, the opportunity of theorizing 5G’s influence on pandemic prevention and control emerges before us. 

### 2.3. Materiality and Affordance

Materiality refers to the stable properties and characteristics of information technology artifacts which do not vary under different environments and times [55]. In terms of 5G, we consider its materiality as the salient features which significantly differ from previous generations of communication and other traditional communication networks, and we do not distinguish the physical materiality which relates to the hardware, and digital materiality which relates to the software [56], because most 5G applications in pandemic control are based on its physical materiality. From this perspective, the materiality of 5G comprises of eMBB, URLLC, and mMTC.

The materiality of IT artifacts is stable and perpetual, however, the environment is volatile and temporary. Therefore, it is hard to explain how the material features of digital technologies enable innovation under dynamic environments [55]. We have to resort to the concept of affordance, which has become the dominant lens to conjecture and explain the possible action that is afforded by the material features of IT artifacts to theorize how 5G enabled pandemic control and prevention. Affordances originating from psychology depict the relationship of the subject to activity and its environment [57]. In the information systems (IS) field, it describes the interactive relationship between the users of IT artifacts and the applied environment [58]. The process of the realization of affordance can be depicted as: (1) users interpret technical objects based on their objectives, which are impacted by the organizational context [59,60]; and (2) the users take action to realize the results that they interpret, since affordance is the potential result of the action [61]. Affordances have been applied in various individual and organizational contexts and Table 2 presents the affordances of typical digital technology.

In this study, the reasons that we employed materiality and affordance as the lens to study how 5G enabled COVID-19 pandemic control are that (1) numerous organizations such as business firms, hospitals, and levels of government used 5G applications because of the unique characteristics of 5G technology; (2) the objectives of the 5G applications were to accomplish the tasks in pandemic prevention and control which were clearly instructed by the China’s Center for Disease Control and Prevention (CDC), meaning the usage of the 5G had a clear objective; and (3) the material characteristics afforded the activities in our study which were manifested as 5G applications, to achieve the objectives. Therefore, it is appropriate that the concepts of materiality and affordance serve as a lens to detect how the material characteristics of 5G afford pandemic prevention and control.

## 3. Research Method

5G is the latest technological innovation, and the COVID-19 pandemic is a very rare public crisis, which offers the opportunity to observe and theorize how 5G enables pandemic prevention and control. Our study selects inductive qualitative research to explore this mechanism [70]. The reasons for why chose qualitative research are as follows: firstly, the mechanism of how 5G enabled the COVID-19 pandemic control is a question of “how”, which can be better solved by qualitative research [71]. Secondly, 5G technology, COVID-19, and 5G-enabled pandemic control are new phenomena, the extant literature cannot explain our question. Therefore, this study is exploratory research, which can be better solved by inductive research. Thirdly, 5G applications play significant roles in many occasions, so we did not distinguish research units strictly in data analysis, rather than viewing the data as a whole to analyze. Generally, inductive qualitative research is an appropriate method to solve our research question.

### 3.1. Site Selection and Data Collection

The research context of this study is the COVID-19 infectious disease, which broke out in the earlier 2020, and had been identified as a pandemic by WHO. We depict the COVID-19 pandemic prevention and control in China as follows:

The first case of COVID-19 occurred in Wuhan on 8 December 2019, and spread rapidly because of the lack of knowledge and protection in the earlier stages. COVID-19 is caused by SARS-CoV-2, a never-seen virus of human beings. The virus spreads mainly through close contact between persons, and it spreads very easily from person to person [22]. To curb the spread of the pandemic, Wuhan took the measure to lockdown the city from 10:00 on 23 January 2020. The city’s public transport was suspended, and the airport, railway stations, and other outlets of Wuhan were temporarily closed. Other provinces and regions in China’s mainland launched the first level response to major public health emergencies and implemented closure management of communities in the most severe stages of the epidemic, which meant that the residents were not allowed to go out except that every family send one member to go out to buy daily necessities every two days. Based on the strict public management, the number of new confirmed infectors and suspected infectors in China’s mainland had been on the decline. On 28 February 2020, the number of cured infectors exceeded the number of confirmed infectors. On 31 March, the number of newly confirmed infectors in China’s mainland was zero in successive days. The National Health Commission of China stated that the spread of the epidemic was blocked. Although there were scattered outbreaks because of the imported cases and goods, the epidemic had not spread wildly in China.

We chose the most severe stage of the pandemic as the research time segment. Firstly, at this stage, the public management was the strictest and the daily life of the residents was restricted, most seriously, offering the typical contexts which are often required in qualitative research. Secondly, the cases in which 5G enabled pandemic control happened in this stage, and the influences were more obvious. 

We collected 243 cases that were related to 5G technology supporting pandemic prevention and control from news and research reports (readers who are interested in the cases could email the corresponding author). These cases mainly introduced how 5G technology played roles in the pandemic prevention and control. Among these cases, 117 were collected and published by China Academy of Information and Communications, and other cases were gleaned from microblog and WeChat. To ensure the authenticity of these cases, we searched them through a search engine. If a case was reported by only one source, we excluded it and retained the cases which were reported twice and more by different sources. Finally, 212 cases were included in our data analysis.

### 3.2. Data Analysis

The data analysis principles abided by the recommendation of Yin [71] and the details of data analysis referred to the process of Sergeeva et al. (2017) [72]. Firstly, according to the three main scenarios of disease prevention and control, namely, diagnosis and treatment of the infectors, cutting off the routes of infection, and logistics support, which were proposed repeatedly in the instructions of the China’s CDC, the cases were classified. Secondly, we summarized what characteristics of 5G played roles in these cases. Thirdly, we cross-compared the results of the first two steps to obtain the features of 5G technology application and the material traits that were applied in each scene, to match the technical features with the scene features, so that we can get the combination mode of technical features and situational features and the results that were generated by the combination. The analysis, as well as the coding that we applied, were conducted in a manner that allowed us to move back and forth between the analysis of empirical data and theorizing, so the steps we describe here are not strictly sequential phases.

Some precautions were conducted to corroborate the findings and guarantee the credibility, transferability, and confirmability, which are important in the trustworthiness of findings in qualitative research [73]. Firstly, the sources, analysts, and researchers were triangulated to guarantee credibility. Secondly, to guarantee the transferability (i.e., the extent to which the interpretation can also be employed in other contexts), we chose as many cases as we could collect before the data analysis. Third, to guarantee the confirmability (i.e., the researchers’ objectivity in interpreting findings), the data were analyzed by the first and second author independently. The results from this analysis and the coding decisions were discussed with coauthors, who contributed to the conceptualization of the findings in terms of a coherent, integrated theoretical scheme.

## 4. Cases Presentation and Data Analysis

According to the WHO’s principles and instructions from China CDC, the primary tasks of pandemic prevention and control can be categorized into three categories: diagnosis and treatment of the infectors, cutting off infectious routes, and logistics support. In this part, we illustrated how 5G enabled these basic tasks.

### 4.1. Diagnosis and Treatment of the Infectors

The diagnosis and treatment of the infected people are the top priority of pandemic prevention and control, which directly determine the life of patients and the prevention of the outbreak again. However, when numerous infected and suspected infected people flooded into the hospitals, the regions with severe epidemic was encountered with the shortage of medical resources, especially doctors and nurses, causing the infectors of COVID-19 not to attain timely diagnosis and treatment.

In the tasks of diagnosis and treatment of the infectors, 5G enabled pandemic prevention and control through innovative medical services. The services included (1) tele-diagnosis of suspected patients, in which the CT and other medical images, nucleic acid test reports of the suspected infectors can be transmitted to the remote hospital through a 5G network, and the remote doctors can diagnose the patients based on these files. Taking a 5G-based CT system as an example, it can directly connect to the hospital’s PACS system, and the raw data that it generated can be transmitted through the 5G network, so that the remote online doctors can “really” see the CT images of patients. In the traditional way, the doctor took the photos of the CT images with a mobile phone camera and sent them to remote doctors. However, the angle of photographing, the light on the screen, and even the color difference that was caused by the camera equipment will cause “distortion” in the definition of the photo, which led the doctor to rarely to make the right diagnosis of the patient, according to the international general DICOM protocol and clinical practice, a set of standard CT images contains 5000 to 6000 consecutive images and many valuable parameters. Therefore, a set of CT files may amount to tens and even hundreds MBs, which is difficult to transmit in real-time through the traditional ways. 5G-based CT systems make the sharing the raw CT images and data in real-time possible, and it supports simultaneous interaction between the people at the on-site and remote sites. Through the 5G-based CT system, doctors in Beijing and Wuhan, 1200 km apart, judged and marked lesions on lung CT images in real-time.

(2) Tele-treatment for the infectors, which is based on the high reliability and low delay, remote experts or medical teams can operate equipment to treat the infectors. Compared with tele-diagnosis, tele-treatment brought new challenges, for example, it demands less information loss and synchronicity of the operational instructions in the transmission between the on-site and the remote sites. The 5G network made tele-treatment for the infectors of COVID-19 become viable, since its traits endowed the network with the capability of transmitting large volume data with high reliability and low delay. The example of Huangpi cabin hospital illustrated tele-treatment well. Due to the lack of medical equipment and resources, the hospital introduced a remote ultrasonic robot which assisted a doctor in Zhejiang Provincial People’s hospital apart from Huangpi about 600 km to treat the patients. Ultrasonic robots with characteristics of real-time and dynamics can generate big volume data and need high performance network to transmit the data. For example, a few minutes of cardiopulmonary ultrasound examination can produce massive ultrasound image data, up to 2 GB. The features of eMBB and URCLL of 5G technology provided a more stable, safer, and faster network to transmit these data. China Telecom’s 5G network ensured the multi-channel and transmission of ultra-high-definition (UHD) and massive data that were required by remote ultrasound. The latency of transmission of instructions decreased to the millisecond level, which can be negligible. Hence, doctors in Zhejiang can treat the infectors in Huangpi based on the data that are generated by the remote ultrasonic robot and transmitted by the 5G network. Table 3 presents more cases in which 5G enabled the diagnosis and treatment of the infectors.

To summarize, in the diagnosis and treatment of the infectors, 5G technology depended on eMBB, URLLC, and mMTC to transmit the UHD images and videos or huge volume of medical data to the remote doctors or experts to make sure that the remote doctors acquired images and videos as vivid as on-site to allow tele-diagnosis. Based on the patient’s condition that was obtained from tele-diagnosis, 5G can ensure the high reliability and super low latency in the transmission, which brings about the synchronization between the on-site and the remote locations, to make the tele-treatment viable. The tele-diagnosis and tele-treatment relieved the unbalance of medical personnel that was caused by the outbreak of COVID-19.

### 4.2. Cutting Off the Infectious Routes

Cutting off the infectious routes is the most critical task in pandemic prevention and control. The first important objective of cutting off the infection routes was to distinguish, screen, and quarantine the infectors, the suspected infectors, and their close contacts. However, it was hard to screen the suspected infectors who were usual with fever from the large scale and fast-moving population via measuring body temperature by mercury thermometer, which was time-consuming and may cause infection because of crowd retention and close contact that was entailed by the retention. Another objective was to discover non-contact ways to accomplish the tasks, which must be accomplished by contact in traditional ways, because the COVID-19 virus can spread via physical contact. The third objective was to supervise the implementation of the lockdown policy under the context of a shortage of workers who were locked at home also.

5G facilitated the achievement of the first objective by screening the floating population and monitoring the home-quarantined people, the second by realizing non-contact operations, and the third by remote supervision. These measures are overlapped to some extent in practice, so we illustrate them by three cases as follows:

5G-based thermal imaging body temperature measurement systems can screen persons with fever from the floating population through non-contact ways. The Haier group integrated far-infrared cameras and infrared rectification instruments to develop a 5G-based non-contact body temperature detectors, which depended on the eMBB of 5G to collect the UHD videos and upload them to the cloud platform through a 5G network. The AI algorithm that was deployed on the platform analyzed the videos to discern the human being’s forehead to measure their temperature. Then, the detector measured the temperature according to the analytical results; if an individual whose temperature was abnormal was detected, the detector alarmed in a timely manner. The security staff would confirm the status of the person. In this way, the potential infector can be detected, and the infectious routes were eliminated to a low extent.

5G-based smart monitors were equipped on the house doors of the home-quarantined people. When the quarantined people opened the door to go out, the monitors alerted and reported to the command center simultaneously, so the potential infectious sources were restricted at home. China Telecom introduced an intelligent monitor, which was called “smart gate magnetic pair terminal”. Based on NB-IoT (narrow band-Internet of things), an innovative technology that was directly deployed on the 5G networks, the terminals monitor the status of doors. If a close contact had to be quarantined at home, the terminals were equipped on her/his door. When the door was open, the terminal alarmed and sent messages to the administrators of community. The terminals had been applied in many communities in Zhejiang, Jiangsu, and other provinces, helping to monitor the home-quarantined people.

5G-based unmanned aerial vehicles (UAV) and robots leveraging 5G’s characteristics of eMBB, URLLC, and mMTC, transmitted UHD images and videos to the command centers of the local government to supervise the implement of the lockdown policy. Qixian County of Hebi City used the UAV as the “sentinel” to supervise the outdoor environment to prevent the residents from loitering, which might cause close contact and transmission of the virus. The UAV took UHD videos and transmitted them to the cloud platform through the 5G network, the AI program analyzed the videos to judge if there was a human being in the videos. If the UAV discovered persons walking around outdoors, it alarmed and commanded them to abide by the lockdown policy. Meanwhile, the UAV transmitted UHD videos to command centers, the staff would handle the complicated situation according to the undistorted videos. Zhejiang, Sichuan, and other provinces took similar measures to supervise the policy of community closure. Table 4 presents the cases in which 5G enabled cutting off the infection routes.

To summarize, in the process of cutting off the infectious routes, based on the eMBB of 5G, (1) a vast amount of data, especially unstructured data, can be transmitted to the cloud platform. AI and BDA programs analyze the data and return the analytical results to the terminals. The terminals perform action according to the instruction, (2) the UHD images and videos can be transmitted to the remote monitors, the monitors get the vivid picture with every detail of the scene and take corresponding actions. 

### 4.3. Logistics Support

Logistics support included three aspects. The first one was to ensure the supply of medical resources and the regular operation of hospitals. The second one was to safeguard the basic living needs of residents that were under lockdown, and the third one was to recover the normal social and economic functions. However, the traits of the infection of the pandemic of COVID-19 caused the logistics to be arduous. The outbreak of COVID-19 encountered the Chinese Spring festival, during which the migrant workers went back to their hometowns, the factories suspended production, reduced the stock, and collected the debts. All of these led to the shortage of medical resources and the difficulty to resumed to manufacture in a short time. The policy of lockdown of neighborhoods commanded the residents not to go out unnecessarily. However, the strict enforcement of the policy brought about a side effect that the essential living requirements of inhabitants were not satisfied, and the normal work and production were interrupted.

The application of 5G ensured the supply of medical resources and management of hospitals, satisfaction of residents’ requirements, and recovery of work and life.

The fact that 5G facilitated the supply of medical resources and the management of hospitals can be indicated by: (1) 5G-based IoT and sensors made the long-distance transportation of precision medical equipment possible, which ensured medical resource supply in the areas with severe pandemic. Sinotrans Group integrated BDA, 5G-based IoT, and cloud computing to develop a management platform called “logistics control tower” to monitor the transportation of high-precision CT machines from Shenyang to Leishenshan hospital. The platform planned the optimal transportation path of the instrument and monitored its vibration and tilt status to ensure its safety. (2) The public supervised the use of medical resources in pressing demand through a 5G-based live broadcast system, since the misappropriation of medical materials had occurred occasionally in the very early stages of the epidemic. Relying on the 5G network and 5G^n^ ultra high-definition live broadcast technology, China Unicom established a 24-h monitoring system of 8-channel streams to assist the supervision of the use of medical materials. (3) 5G-based robots, such as a 5G-based medical assistance robot, 5G-based pill delivery robot, 5G-based sterilizing and cleaning robots, and so on assisted medical staff to complete remote medical care, body temperature measurement, disinfection, cleaning, and other work.

5G-based technologies can support the basic needs of home-locked residents through many ways: (1) 5G-based driverless vehicles and unmanned distribution robots delivered foods and other necessaries of life to the people who were under lockdown. China Unicom and Suzhou Changxing Co. (Suzhou, China) together launched “5G-based unmanned logistics distribution vehicle”. The vehicles delivered the order within the radius of 3 km to ensure the basic living needs of the residents without person-to-person contact. (2) 5G applications provided people with more diverse and vivid entertainment to meet their spiritual needs. For example, the construction team of Leishenshan hospital opened the 24-h continuous 4K HD, 360° VR live broadcast. More than tens of million audiences watched live to eliminate the loneliness that was entailed by the lockdown and empathize how to fight the epidemic to increase their enthusiasm to participate in the fight against COVID-19. (3) 5G-based telemedicine provided medical services to people who could not go to hospital because of the lockdown. Netease Yunxin Co. (Hangzhou, China) and 1.0 Flowed Medical company developed 5G-based tele-consultation systems to provide remote medical services for chronic and non-emergency patients under lockdown.

5G-based remote real-time video conference systems, telecommuting systems, and distance learning systems provided users with an immersive experience to realize the recovery of work and production to a certain extent. These 5G-based systems relying on the eMBB, and URLLC transmitted UHD images, videos with every detail to users, and instructions with ultra-low latency, which can smooth the interval between the stages of tasks enhance the experience of remote education and work. For example, during the epidemic, Huawei Digital Town in Qinzhou, Guangxi, used the 5G-based conference system that was built by China Telecom to sign contracts of investment projects online. Through cloud computing, 5G-based robot arms and other technologies and representatives that were scattered in different places were gathered virtually. Every party signed on the mobile phone that was installed with signature software, and the 5G robot arms on site synchronously wrote the signature on the contract text. Table 5 presents the cases in which 5G enabled logistics support.

## 5. Theory Building

This section presents the model of how 5G enabled the prevention and control of the pandemic of COVID-19 that is based on our analysis of the cases. We grounded two types of key actions that were afforded by 5G technology: de-spatialization and spatialization, and the process of coding is presented in Table 6. Spatialization provides non-contact ways to complete the tasks which are supposed to be completed in contact, and de-spatialization provides remote operations to complete the tasks which are supposed to be completed on-site. Spatialization facilitates the pandemic prevention and control by cutting the transmit routine, and de-spatialization enables the pandemic prevention and control by realizing telemedicine.

Both types of actions are enabled by the material features of 5G technology in terms of eMBB, URLLC, and mMTC. The model is visualized in Figure 2, and in as follows, we provide a general overview of the model and then depict its key components in terms of (1) the main objectives of pandemic control and prevention, (2) de-spatialization, (3) spatialization, and (4) the comparison between de-spatialization and spatialization. 

### 5.1. The Main Objectives of Pandemic Control and Prevention

The outbreak of extremely infectious epidemics lead to the shortage of medical resources in the areas with the severe epidemic, especially doctors and nurses. The channels of infection, including close contact and respiratory, meant that avoiding close contact became the measure to stop the viral spread. It can be found that the measures that were taken by the Chinese government were to solve the shortage of medical staff in the worst-hit regions by appropriating medical staff from areas with light contagion and to minimize the person-to-person contact by locking down the city and calling for people to wear a mask, and all of them should be supported by abundant logistics supply. Hence, the objectives of the diagnosis and treatment of infectors, cutting off the infection routes, and logistics support were transformed into eliminating the defect in medical resources and completing tasks without close contact. For example, the lockdown of communities to cut off the infectious routes meant that the basic requirements of residents can’t be met in a timely manner, and the solution can be boiled down to distributing subsistence goods in a non-contact way. The unbalance of medical staff between the severe and light areas can be solved by the appropriation of medical staff. However large-scale population movement might cause infection, and too many staff gathered in one area might exceed its reception capacity, especially under the situation of closed management. Therefore, the task of balancing the medical staff fell on telemedicine, including tele-diagnosis and tele-treatment, which had extremely high requirements for the resolution of images and videos and the synchronization of the operation of the remote and on-site. Therefore, the practical objectives were transformed into technical objectives.

### 5.2. De-Spatialization

eMBB of 5G offers the network with the capability of transmitting the UHD images and videos to the remote location, making the remote site get the scene as vivid as on-site. URLLC of 5G offers the network with the capability of eliminating the information loss and delay of transmission, making the operation of the on-site be synchronous with the remote. The vivid scene and synchronous operation dispel the information distortion, the delay of operation, and bad experience, which were caused by the long distance between the on-site and the remote, providing the remote lifelike experience to complete the tasks. All of these seem to eliminate the space between the on-site and the terminal, so we called it as de-spatialization. 

The context in which de-spatialization works includes telemedicine, which removes the unbalance in the medical staff that is caused by the outbreak epidemic and tele-entertainment and telecommuting, which help to resume regular life and work that were interrupted by the epidemic. Telemedicine is categorized into tele-diagnosis/tele-consultation and tele-treatment. 4 K, even 8 K images and videos can reduce blurring and information loss that is caused by image compression and ensure that remote doctors can observe every detail to diagnose the patients. As for tele-treatment, URLLC ensures that the remote operation can be synchronized to the on-site location. For example, in tele-surgery, if there is a delay in the transmission of instructions, the amount of bleeding will increase to endanger the life of patients. 5G technology can shorten the delay between the terminal and the onsite to no more than 1 millisecond, which is completely comparable to the field operation. Tele-entertainment and telecommuting highlight the experience, which can be ameliorated by the 5G network. The vivid images and videos and the synchronous operations make the work experience smooth and the entertainment experience can be immersive.

### 5.3. Spatialization

5G’s enhanced bandwidth can (1) transmit a huge volume of data, especially unstructured data that were collected by the terminal sensors to the cloud platform in real-time. The data are analyzed to form instructions and returns them to the terminal, which will perform operations according to the instructions; (2) transmit the on-site high-definition images and videos (or trigger events) to the remote command center in real-time, and the staff make a judgment according to the images and videos (or trigger events) and take the corresponding action. All these applications look similar to generating a space between the parties of the tasks to make the contact be non-contact, hence we named it as spatialization.

The context in which spatialization works includes the cutting of infectious routes such as non-contact operations in body temperature measurements and the unmanned distribution of living substances and remote monitors such as inspection of the implementation of lockdown policies. The 5G network with features of enhanced bandwidth and low latency transmits the massive data that are collected by the terminal sensors to the cloud in real-time. The AI and BDA programs that are deployed on the cloud platform analyze the data, form instructions based on the analytical results, return the instructions to the terminals, and the terminals operate according to the instructions. For example, in the non-contact people screening system based on 5G, the terminal camera transmits the face image (with mask) to the cloud platform in real-time. The AI and BDA program recognize the face and compare it with the face images in the database to detect whether she/he stayed or passed by an epidemic area. The feature of mMTC ensures that the performance of the devices will not be affected when there are an immense number of terminals in the network. For example, the cabin Hospital of Wuhan Optics Valley Science and Technology Center established a 5G-based Nb-IoT/Lora network. Every infector wore a bracelet with an embedded IoT module to detect his/her movement. When the patient overstepped the isolation area, it alarmed. The cabin hospital had a maximum of 850 patients, and with other sensors, up to tens of thousands of sensors were connected to the network at the same time. The traditional 4G and WiFi technologies couldn’t ensure that such a large number of terminals could be accessed to the network in a small area without mutual interference. The 5G network had the capability of keeping the normal operation of the devices while many devices access to the network at the same time.

### 5.4. The Comparison between De-Spatialization and Spatialization

De-spatialization and spatialization seem alike to some extent, since both of them make tasks accomplishable without close contact. However, there are essential differences between them, which we depict as follows:

Firstly, the purpose and mechanism of them are different. The purpose of de-spatialization is to eliminate the information loss and latency in the transmission. Limited to bandwidth, 4G and WiFi have to compress the files before transmitting huge volumes data and HD images and videos, which causes information loss; and limited to the network structure, the latency of 4G and WiFi are usually high which cannot meet the requirement of tele-treatment. Therefore, the mechanism of de-spatialization is that the eMBB of 5G technology transmits high-definition images and videos and huge volume data and URLLC reduces the delay and data loss in transmission, giving remote operators the same experience as the onsite to complete the task. The purpose of spatialization is to eliminate the close contact of persons. Its mechanism is that the 5G network transmits a large amount of data to the cloud for analysis, returns the analytical instructions, and makes the terminals carry out the operation according to the instructions to avoid contact between people.

Secondly, the distance which de-spatialization and spatialization work is different. The distance of de-spatialization is usually far, which may be measured in tens of kilometers at least. The reason is that the distance between parties of remote operation is long, for example, the remote doctors and on-site patients in tele-diagnosis and tele-treatment may scatter in different cities; tele-commuting may be through half city. The longer the distance between the parties of tasks, the more severe the delay of transmission and information loss are, which caused the bad experience of tele-operation. Therefore, the contexts where de-spatialization plays a role are distant parties. The distance of spatialization is usually proximate, which can be measured in meters, or hundreds of meters at most. That is because the infectious routines of the COVID-19 virus are respiratory and close contact, so it is enough that spatialization establishes the social distance between persons.

Thirdly, another significant difference between de-spatialization and spatialization is their influence on automation. An important result of spatialization is the unmanned operation, such as the unmanned distribution of living materials. In spatialization, people’s participation is eliminated to avoid close contact. In de-spatialization, people’s participation will not be excluded. For example, in tele-surgery, the operators of the surgery are still remote doctors rather than medical machines or robots.

## 6. Theoretical Contribution

Grounded in the material features of 5G technologies, i.e., eMBB, URLLC, and mMTC, this study presents a theoretical model of how 5G enables COVID-19 prevention and control, which extends on prior work on digital technologies enabled pandemic control by explaining how spatialization yields non-contact ways to accomplish tasks which have to been done by close contact and how de-spatialization yields remote ways to accomplish the tasks which have to been done on-site. In this section, we discuss how our model contributes to the literature on digital pandemic control and to the literature on IS.

### 6.1. Contribution to Research on IS Scholarship

5G technology is one of the latest digital innovations and even hasn’t been commercialized in a large scale; however, it has indicated extreme potential in many aspects. Therefore, to theorize how 5G enabled pandemic prevention and control has significant theoretical insights.

We employed the concept of affordance to explain how 5G enabled pandemic control. Affordance is the intersection of technology and its specific applying environment. Novel and creative applications of digital technologies will emerge along with the changes in the use context. In COVID-19 containment, the specific forms of application of 5G varied according to the context. However, what explains the similarities in the occurrence of pandemic control across contexts? Therefore, it has to advance to induce the reality to theorize how such applications occurred. Our study focused on how and why 5G brings about influence on pandemic prevention and control by diverse applications under different contexts, for instance, why telehealth and driverless technology, which are seemingly unsimilar technologies can enable pandemic control. Grounded from 212 cases which applied 5G to combat against the COVID-19 pandemic, we induced two kinds of affordances of 5G that play roles in this combat: spatialization and de-spatialization.

These two kinds of affordances offer us new insights to understand how 5G produces social influences in the stage where most cognate research on 5G mainly concentrate on its technological traits. 

Firstly, the majority of extant literature on how 5G enabled pandemic prevention and control was concerned with the phenomena themselves and ignored how and why they happened, which meant that they cannot offer theoretical insights on this topic. Our study leveraged the concepts of materiality and affordance to analyze 212 cases in which 5G played important roles and discerned two basic ways that were afforded by 5G, which are spatialization and de-spatialization. These two ways uncover the nature of the role of 5G in pandemic prevention and control. 

Secondly, the findings of de-spatialization contribute to related research on the management of distance, which has been a focus of scholars for decades [74,75]. Parties with near distance can be interacted better than the parties with long distance, just as Tobler (1970) [76] said “everything is related to everything else, but near things are more related than distant things.” However, increasingly organizations have adopted distributed work meaning that the side effects (also known as distance decay) that are brought by distance can’t be negligible. The emergency of the Internet relieved the distance decay. Diverse definitions and theories had been proposed to explore how to achieve tasks under geographic distance. Cairncross and Cairncross (1997) [77] offered the concept of “death of distance” to depict the Internet as the aggregation of virtual technology that can reduce the information transmission cost, the marketing cost of digital products, and the communication cost, and other costs that are introduced by geographical distance. Other similar concepts included “death of geography” [78], “end of geography” [79], and “friction of distance” [74], whose main point is that the geographical distance does not matter in cyberspace since people living in faraway towns were enabled the capability to interacting with other people by the Internet. The theories which were used to explain the friction of distance included flow theory [80,81], social presence theory [82], and media richness theory [83,84]. The experience of flow is affected by the quality of information and network [85] and promotes the users’ satisfaction of distance learning [86]. Synchronous communication [87] and video conferences [56] are more likely to bring about social presence to improve the results of distance education [88] and online coordination and cooperation [89]. Media richness theory points out that different media loads different quantity and quality of information, which determines the results of communication and media with more richness offering the experience of flow and social presence [90].

On the one hand, de-spatialization inherits the main insights of cognate research on distance, for example, the UHD images and videos, and ultra-low latency between the on-site and the terminal provides rich media to bring about immersive experience, lifelike social presence to improve the effect of distance learning, telecommuting. On the other hand, de-spatialization modifies the conclusion of prior research, for example, media richness theory suggests that face-to-face is the best way to communicate since it can provide the most detail of parties in the chat [90]. However, 5G-based de-spatialization can achieve equal and even beyond the results of face-to-face, which strengthens the results of Walther’s research [91]. Through offering more experience of flow and social presence, it realized tele-operation, such as telemedicine, telework, and distance education. There are scholars deeming that the distance matters still in the age of Internet [92,93], and this study proved that 5G-based de-spatialization can enhance the distance decay through its characteristics.

Thirdly, spatialization is a very situational definition in pandemic control, which is the converse of de-spatialization. In most work contexts, eliminating the distance is the main objective to improve efficiency, however, the infectious trait of contact with COVID-19 demands the avoidance of close contact. We grounded the concept of spatialization from 212 cases. This novel definition offers new insights on extant phenomena. For example, automation including driverless vehicles and robots are to improve efficiency and remove the negative effect that is caused by the shortage of human resources in pandemic prevention and control, their objectives are to avoid contact. The de-spatialization has generality to explain other contexts, such as in the processing of hazardous substances, the usability of robots that not only improves the efficiency, and more importantly is to avoid the worker contact with the substance.

Fourthly, through reviewing the literature on 5G, including the Basket of Eight journals and other journals that are major in information systems, we found that there were rare studies on 5G’s business and social values. However, 5G, which is viewed as one of the most disruptive innovations, can bring about a revolutionary change in business and society. The reasons that 5G’s business and social value has not received enough attention are that (1) lot of challenges exist in 5G technology since it is still in its infancy, for example, there has not been a consensus which network structure is better SA or NSA, so the studies on 5G mainly focus on technological aspects; (2) there are no abundant application scenarios for 5G meaning that it has not been used commercially. For example, the immersive experience has not yet become mainstream, so the daily requirement of network can be met by 4G and WiFi; and the fiber-optical network can meet the business and social demands of the network, so scholars have paid little attention on 5G’s business and social values. However, this study indicates that 5G is an influential weapon to combat the COVID-19 epidemic, for example, telemedical consultation is a regular way that many chronic patients will take, but limited to network conditions, the effect is not satisfactory. 5G-based telemedicine improves the accuracy of consultation through de-spatialization. Therefore, another contribution of our study is to call for scholars to pay attention to 5G’s social and business values through indicating the cases and grounding the means of how 5G enabled the pandemic prevention and control.

### 6.2. Contribution to Digital Pandemic Control

At a general level, we discovered that 5G allowed people to accomplish tasks in a non-contact way, which was supposed to be accomplished by close contact, and to achieve tasks in remote locations which were supposed to be achieved on-site. These findings of our study offer insights into pandemic control, especially the digital-enabled pandemic control.

Firstly, 5G technology changes the diagnosis and treatment of the infectors. The outbreak of the epidemic and the lockdown it entailed caused an extreme unbalance in medical staff and resources. The existing telemedicine relies on optical fiber to connect the remote location and on-site, however, the newly built hospitals are hard to lay the optical fiber network. The 5G network is easy to be deployed on establish telemedical systems to enable the diagnosis and treatment of the infectors. On the other hand, during the period of lockdown, the mental health of the public is also affected because of the boring life. With 5G-based VR/AR technologies, quarantined people at home can obtain an immersive entertainment experience and relieve psychological trauma, which is a typical case of 5G-enabled psychotherapy. The de-spatialization offers a theoretical explanation for these measures of pandemic prevention to contributes to the literature on pandemic control.

Secondly, digital technologies, especially 5G technology, enabled the cutting off of the infectious routes. The important measures of cutting the transmission of infectious diseases are “detection, isolation, and tracking”, which are suggested by WHO. In the earlier outbreak, detecting the infectors can effectively prevent the epidemic. However, the research and production of the test toolkits are lagging. Therefore, if large-scale lockdown is carried out, the epidemic can be stopped. Yet the complexity of social systems makes it difficult to implement lockdown. 5G technology can realize tele-commuting and automatic production by de-spatialization, offering the possibility to carry out large-scale isolation to a certain extent. Spatialization offers home-quarantined people in a noncontact way to complete the task, which reduces the inconvenience that is caused by pandemic prevention and control. Therefore, 5G technology enables the measures of epidemic prevention and control, which is contribution to the literature on this topic. 

Generally, the influence of the 5G exerted in COVID-19 containment is theorized into spatialization which eliminated the infectious routines, and de-spatialization which facilitated remote operations, both of which are contributions to pandemic prevention and control.

## 7. Practical Implication

Our study emphasized that how 5G technology enabled pandemic control and thus offers practical implications.

Firstly, the winter caused the outbreak of COVID-19 again, even in the area where the epidemic is well controlled, sporadic infectors still emerge. The pandemic control has become part of regular social life, which can obtain insight from our study. For example, if there are sporadic case outbreaks in certain areas, they can quickly deploy the relevant equipment based on 5G, such as de-spatialization equipment to deliver resources to people in lockdown, and spatialization equipment to screen floating people, to hinder the spread of the epidemic.

Secondly, the pandemic of infectious disease is a disaster for the human being, even in the modern social society because of the lagging of the development of vaccines and medicine; the earlier stages of the outbreak may cause huge casualties. Therefore, for respiratory and contact infectious diseases, quarantining the infectors and lockdown communities are still an efficient measure to control the pandemic. The findings of this research can shed light on how to facilitate the pandemic control.

Thirdly, the generality of spatialization and de-spatialization makes it possible that 5G enables other scenarios. For example, in the treatment of toxic waste, it is necessary to complete the task in a non-contact way. Therefore, the control measures can be formulated from the perspective of de-spatialization.

## 8. Limitation and Future Direction

The limitations in our study lie in firstly, 5G is still in its infancy and has not been used universally in the whole society, and its application needs to be based on solid infrastructure construction. Therefore, that 5G enabled pandemic control still occurs in a small range, so its impact may not be as common as the extent that our study indicated.

Secondly, 5G enabled epidemic prevention and control also needs to be combined with other digital technologies, such as big data analysis, cloud computing, and robots etc., but our study does not consider the impact of these technologies. In the future work, we will consider how these technologies enable epidemic prevention and control.

## 9. Conclusions

The prevention and control of COVID-19, on one hand, demands people to keep distance to avoid the infection of the virus. On the other hand, it has to cooperate closely to clear the highly contagious disease. 5G plays important roles in solving these contradictory tasks. Therefore, this study explored how 5G enabled the COVID-19 pandemic prevention and control based on the conceptions of affordance. Through the analysis of 212 cases, the approaches of 5G facilitating the pandemic prevention and control were theorized into spatialization and de-spatialization. Spatialization provides non-contact ways to complete the tasks which are supposed to be completed in person, and de-spatialization provides remote operations to complete the tasks which are supposed to be completed on-site. This study makes theoretical contributions to digital pandemic prevention and control, and the literature on 5G, also offers practical implications.

## Figures and Tables

**Figure 1 ijerph-19-08965-f001:**
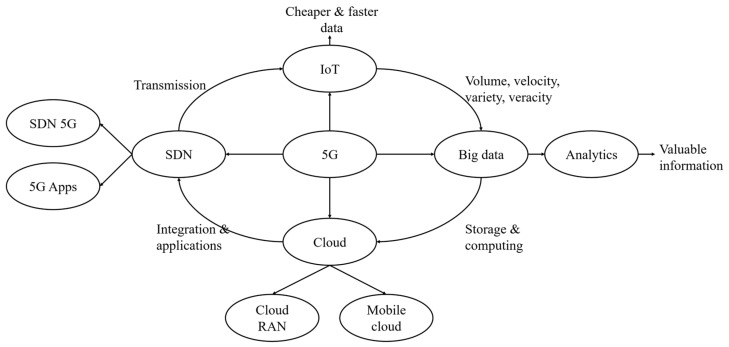
How 5G enables other digital technologies. Source: adapted from Lin, Lin & Tung’s research [33].

**Figure 2 ijerph-19-08965-f002:**
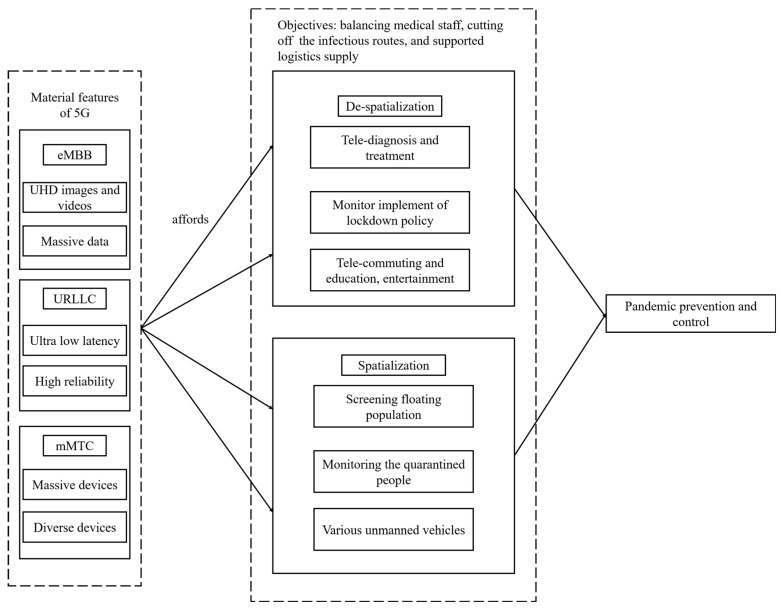
The model of 5G that enabled the pandemic prevention and control.

**Table 1 ijerph-19-08965-t001:** Application Scenarios of 5G.

Application Scenarios of 5G	Description
Smart healthcare	Providing healthcare services through smart gadgets (e.g., smartphones, smartwatch, wireless smart glucometer, wireless blood pressure monitor) and networks (e.g., Body area network, wireless local area network, extensive area network) [34], including telehealth [35], Robotic telesurgery [36], and telehabilitation [37], and so on.
Industry 4.0	Monitoring and tracking of machine and processes, metering of resource, and telecontrol operations for better understanding to the readers [38], including Internet of Vehicles [39], smart logistics [40], smart manufacturer [41], teleoperation [42,43].
VR/AR	Changing the perception of a person about the objects in the context of visual and audio environment [38], including 5G-based VR/AR game [44], smart education [45], sports [46,47].

**Table 2 ijerph-19-08965-t002:** Affordances of typical digital technologies.

Research	IT Artifacts	Types	Affordances
Zammuto et al. [62]	Information technology	Definition development	Visualizing entire work processes, real-time/flexible product and service creation, virtual collaboration, and mass collaboration, simulation
Treem & Leonardi [63]	Social media	Review	Visibility, editability, persistence, and association
Dong et al. [64]	Social commerce	Definition development	Visibility, meta voicing, triggered attending, guidance shopping, social connecting, and trading
Shin [65]	VR	definition development and hypothesis testing	Presence affordance, immersion affordance, comfortability affordance, empathy affordance, and embodiment affordance
Krancher et al. [66]	PaaS	Definition development	Shaping environment, reusing software service, self-organizing, and triggering continuous feedback
Achmat and Brown [67]	AI	Review	Automate business processes, customize end user interaction, proactively anticipate and react to changes, augment and upskill the workforce, assist decision making, improve risk management, and develop and enhance intellectual property
Naik et al. [68]	IoT	Definition development	Improve business processes, manage stock, reduce costs, and provide transparent data access
Dremel et al. [69]	BDA	Definition development	Establishing customer-centric marketing, provisioning data-driven services, data-driven developing, and optimizing production processes

**Table 3 ijerph-19-08965-t003:** The cases in which 5G enabled the diagnosis and treatment of the infectors.

Application	Cases
Tele-diagnosis	Medical teams in Beijing offered tele-consultancy for the infectors in Wuhan through 5G-based telemedical systems.
	China Unicom’s 5G network and cloud platform supported national telemedical teams to bring together the experts to conduct tele-consultancy for the infectors in Hubei province.
	Hisense’s 5G-based telemedical system provided tongue coating imaging with high fidelity, remote auscultation of lung sounds, making the traditional Chinese medicine diagnoses of patients in distance possible.
Tele-treatment	Ambulance connecting into 5G network and equipped with 4K cameras can transmit HD images and videos of the infectors to the hospital in time, so the doctors can formulate a treatment plan in advance.
	Leveraging the 5G network supplied by China Telecom, doctors in Zhejiang manipulated the mechanical arms to take operation of pulmonary drainage on the infectors in the Wuhan.
	Shanghai Chest Hospital conducted remote bronchoalveolar lavage operation on the patients, in which the operation was done by robot on-site and the robots were controlled by remote medical staff. In these processes, the remote and the on-site were connected by the 5G network.

**Table 4 ijerph-19-08965-t004:** The cases in which 5G enabled cutting off the infection routes.

Applications	Cases
Hospital management	Butel company launched 5G-based remote prevention and control system to assist Chaoyang hospital to monitor the quarantined wards. Doctors out of the quarantined areas can observed the infectors based on the HD and uninterrupted all-day videos transmitted by 5G network.
	First affiliated hospital of Kunming medical university launched visit systems, which offered the visitors of the infectors with two-way and real time transmission of HD audio and video, making the visitors watch every detail of the quarantined patients.
	The affiliated hospital of Qingdao University developed ward systems based on 5G video. The systems comprised of AR glasses for doctors, high-definition remote video interactive system, 5G panoramic VR real-time display system, etc. which used the characteristics of the 5G network to realize the real-time sharing of multi-media documents and provide the “experience like on the scene”.
Management of lockdown	5G-based intelligent service robots guided patients and propagated knowledge of epidemic prevention in the hospital hall where there was the densest flow of people. The usability of the robots avoided close contact between people, reduced the spread of the virus, and made the epidemic prevention policy be rooted in the public, and assisted the epidemic prevention and control. These service robots had been applied in many cities, such as Hangzhou, Jinan, and so on.
	In Zhejiang province, the police UAV team had flown more than 8000 sorties, covering more than 5000 villages and communities, warning the gathered people more than 20,000 times, and reminding more than 13,000 people to wear masks.
Screening the floating population	5G-based thermal imaging of human body temperature measurement equipment was widely used in subways, railway stations, hospitals, schools, government, enterprises, shopping malls, stores, dormitories.
	Koland developed smart tracing system. The system collected HD photos of passengers and transmitted them to a cloud platform through the 5G network, the platform analyzed the data to confirm the passengers if he/she had been in an area with severe epidemic.

**Table 5 ijerph-19-08965-t005:** The cases in which the 5G enabled logistics support.

Applications	Cases
Supply of medical resource	Haier Group deployed 5G-based industrial Internet to promote the exchange of the information of raw materials of medical resources to resume the production of masks and other products.
	Beijing Ditan Hospital, Wuhan Huangpi Hospital, and many other hospitals employed 5G-based robots to achieve medical care, disinfection, cleaning in non-contact ways.
	Guangdong provincial people’s hospital applied 5G-based delivery robots to distribute the medical resources in point-to-point ways.
satisfaction of residents’ requirements	JD group deployed robots and unmanned vehicles in Wuhan to deliver the life necessaries to home-quarantined residents in non-contact ways.
	“Yat-sen 5G-Based internet hospital” provided high-definition, no delay video consultation, which met the basic medical requirements of home-quarantined people and avoided the potential contagion because of gathering in the hospital.
	China Unicom launched 5G interactive live broadcast of a special topic of Hubei tourist attractions. The broadcast offered people to enjoy the scenery online with an immersive experience.
recovery of the work and life	Guizhou Provincial People’s Congress adopted the videoconferencing system-based China Mobile’s 5G network to hold remote meetings. The ultimate low delay and HD videos smoothed the process of the meeting, providing the experience rivaling with offline conference.
	Schools in Anhui Province used a mobile app called Wanxin campus to carry out online education. Through the 5G network, the app provided audio and video, interactive whiteboard, interactive live broadcast, and other teaching methods, to achieve a more authentic teaching experience for the teachers and students.
	State Grid Hangzhou Electric Power Company designed 5G-based inspection robots for cable tunnels. The robots inspected the tunnel for 24 km in 40 days, to avoid the infection of COVID-19 because the tunnels were an enclosed space where it was easy for virus to spread.

**Table 6 ijerph-19-08965-t006:** The process of the coding of de-spatialization and spatialization.

Themes	The Results	Cases
De-spatialization	blurring images and videos → UHD images and videos	1. The Department of radiology West China Hospital used 5G network and remote CT scanner to perform medical diagnosis for patients in Ganzi, 300 km away from the hospital. 2. Hangzhou, Wuhan, and Jingmen carried out tele-diagnosis and treatment by the 5G network, bringing together experts from the Sir Run Run Shaw Hospital in Hubei to jointly discuss the critical cases of COVID-19. 3. Jianggan District integrated 5G technology and UAV which were equipped with ultra-high-definition cameras to examine the implementation of lockdown policy. The UAV transmits 4K videos to the command centers through the 5G network, and the staff took actions based on the HD videos.
	high latency and low reliability → ultra-low latency and high reliability	1. Doctors in Shanghai People’s Hospital conducted operation on the patients in Wuhan. The robots in Wuhan were connected with doctors in Shanghai by the 5G network to eliminate the delay between the on-site and the remote which was caused by the distance. 2. The UAVs which were used to monitor the lockdown policy uploaded the videos in real-time to the command center, the center took actions immediately.
	bad experience → immersive experience	1. China Mobile provided online education systems based on 5G to the school in Xinxiang city. The systems integrated multi-teaching methods to imitate offline education in the classroom. 2. Department of telemedicine of Chinese PLA general hospital offered remote training of the prevention and control of COVID-19 to other PLA hospitals that were scattered all over the country through remote training systems based on 5G, which offered every detail of the training. 3. Using 5G technology, the ambulance performed the testing as soon as it received the patient and transmitted the results to the hospital immediately; during the treatment in the hospital, 5G-based equipment realized tele-consultations and analyzed the patient’s condition, and even performed remote surgery; use remote video to track the patient’s recovery after the hospital, thereby reducing misdiagnosis caused by poor communication and smoothing the diagnosis and treatment.
Spatialization	small data → massive data	1. China Unicom and Meituan jointly developed 5G-based driverless delivery vehicles to deliver the orders. The 5G network ensure the transmission of huge volume data required by the vehicles. 2. 5G-based automated guided vehicle (AGV) assisted the management and control of production process by machine vision technology which depended the transmission of big data through the 5G network. 3. Didi group developed 5G-based video systems to detect whether the passengers wore facemasks based on analytical results of unstructured data transmitted to cloud platform by the 5G network.
	delay in transmission → synchronization in transmission	1. 5G ultra-low latency met the communication requirements of equipment in the factory, provided remote control for the intelligent production, helping to resume regular running of the manufacturer. 2. For the transportation of emergency materials, the remote command room obtained real-time information through the 5G network about the vehicles hundreds of kilometers away and its surrounding environment, and sent instructions such as start, acceleration, deceleration, and steering to control the vehicles. These vehicles had been applied in the transportation of contagious materials in Wuhan and other cities. 3. Hangzhou TV station and Hangzhou branch of China Unicom have carried out a 24-h live broadcast of donation by Red Cross Society for the public to supervise the condition of charitable materials in real-time.
	accessibly of a few devices → accessibly of massive and diverse devices	1. 5G intelligent robot systems include 5G-based medical assistant robots, 5G-based disinfection and cleaning robots, 5G-based medicine delivery service robots, 5G-based temperature measurement inspection robots, and other kinds of 5G-based robots were connected simultaneously into the 5G network in the Shanghai Sixth People’s Hospital to assist the management of the infectors. 2. Leishenshan hospital realized Gigabit network coverage and can receive 5G signals, which can carry the concurrent communication requirements of 25,000 people and meet the network needs of remote command, tele-consultation, telesurgery, and data transmission.

Note: the “→” means the change which is brought about by the 5G network.

## Data Availability

Data can be obtained by emailing the corresponding author or downloaded at the website (https://pan.baidu.com/s/13FyIYEK3DkTa-lyexmHSMw?pwd=lsv5 (accessed on 1 April 2020)).

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
