# Peer review of "How the 5G Enabled the COVID-19 Pandemic Prevention and Control: Materiality, Affordance, and (De-)Spatialization"

_ijerph, 2022, doi:10.3390/ijerph19158965_

Round 1

Reviewer 1 Report

In this paper, Authors studied how the 5G enabled the COVID-19 pandemic prevention and control in terms of materiality, affordance, and (de-)spatialization.

The topic is interesting and shows how the technology in general and 5G in specific were important elements in controlling pandemics. The paper is well written and presented. However, there some suggestions to improve the contribution of this paper.

- 5G is not stand-alone technology in terms of prevention and control of the pandemic. It can be named a facilitating technology that makes other important technologies more active and easier to function. Artificial Intelligence played the most important role in controlling and preventing the pandemic through machine vision as an example. Therefore, authors should highlight how 5G facilitates such technologies in order to make the picture more clear.

- Robotics also was a fundamental technology that should b also considered.

- Authors should add a section to analyze the relationship between these technologies and the 5G for controlling and preventing the pandemic.

- How about existing technologies before 5G? such as 4G in most countries? Authors should study the utilization of the 4G and other older technologies as well in the countries that are still using it. 

Reviewer 2 Report

The work makes an interesting study on the applicability of 5G technology in preventing and controlling COVID-19. 

The authors conducted qualitative research on how 5G enabled the COVID-19 pandemic control. However, there is little data in the proposed work that can help with a quantitative analysis. 

It would be interesting to do a quantitative analysis using simulations based on collected data. In this way, the results would provide the reader with a better understanding of the results. 

Including a proposal for a quantitative analysis based on the data collected would be interesting. In addition, a comparative analysis between the use of 5g and strategies that use conventional networks would be interesting. This way, the cost and gain of using 5G could be analyzed compared to other conventional networks. 

In line 207-208. Indicate the references to the sites covered in the sentence. 

In line 175. Please provide more information regarding the “IS field”. 

Please rewrite this sentence: “AI technology enabled the pandemic prevention and control [26] by predicting the spread of the disease [27], and assisting to screen drugs can cure the COVID-19 from existing drugs [28, 29].” 

Please rewrite this sentence:Related studies mainly focus on the engineering requirements of the application scenarios, and little research focuses on the business and social value of 5G [48, 49], which leads to the literature on 5G are mainly published in the journal of engineering and technique.” 

Please rewrite this sentence: “Through the analysis of the literature on 5G and its impact on pandemic prevention and control, on the one hand, because 5G has not been used commercially, so research on it mainly focuses on the possible application scenarios and their engineering requirements, the business and social value is neglected.” 

Please rewrite this sentence: "However, the materiality of IT artifacts is stable and perpetual, so it is hard to explain how the material features of digital technologies enable innovation under diverse environments.” 

Please rewrite this sentence: “Therefore, this study is an exploratory research, which is can be better solved by inductive research.”

Reviewer 3 Report

This is an important paper. You have described a good number of cases.

It reads well, but I feel the paper could be improved:

1) Check the tables, can the information be condensed?

The theory section 5 needs a paragraph or two  in the beginning. Please describe spatialisation and and de-spatialisation in more detail. What do you mean and why this is important? 

There is a lot of theoretical information which could be taken from later sections and re-written earlier on. 

remove section 5.1
